

# Effect of anthropogenic aerosol emissions on precipitation in warm conveyor belts in the western North Pacific in winter – a model study with ECHAM6-HAM

Hanna Joos[1], Erica Madonna[1,2], Kasja Witlox[1,3], Sylvaine Ferrachat[1], Heini Wernli[1], and Ulrike Lohmann[1]

[1]ETH Zurich, Institute for Atmospheric and Climate Science, Zurich, Switzerland
[2]Geophysical Institute, University of Bergen and Bjerknes Centre for Climate Research, Bergen, Norway
[3]Zurich Insurance Company Ltd, Zurich, Switzerland

*Correspondence to:* H. Joos (hanna.joos@env.ethz.ch)

**Abstract.** While there is a clear impact of aerosol particles on the radiation balance, whether and how aerosol particles influence precipitation is controversial. Here we use the ECHAM6-HAM global climate model coupled to an aerosol module to analyse whether an impact of anthropogenic aerosol particles on the timing and the amount of precipitation from warm conveyor belts in low pressure
systems in the winter time North Pacific can be detected. We conclude that while polluted warm conveyor belt trajectories start with 5-10 times higher black carbon concentrations, the overall amount of precipitation is comparable in pre-industrial and present-day conditions. Precipitation formation is however supressed in the most polluted warm conveyor belt trajectories.

## 1 Introduction

The interaction of aerosol particles with clouds and radiation is the largest source of uncertainty for estimating the total anthropogenic forcing since pre-industrial times (Boucher et al., 2013). Anthropogenic aerosol particles such as sulfate and carbonaceous aerosols have substantially increased the global mean burden of aerosol particles from pre-industrial times to the present-day. While the largest increases in aerosol emissions in the 20th century were located in Europe and North Amer-
ica, nowadays anthropogenic aerosol emissions are highest in Southeast Asia (Granier et al., 2011; Klimont et al., 2013).

Aerosol particles affect the vertical extent, lifetime, phase and optical properties of clouds by acting as cloud condensation or ice nuclei. Whether aerosol particles also impact precipitation and if so how, is controversial. The scientific review of aerosol pollution impact on precipitation by Levin



and Cotton (2009a) concluded that clear causal relationships between an increase in aerosol particles and changes in precipitation are difficult to identify and even the sign of the change in precipitation is uncertain. Most studies (both observational and numerical) have analysed the possible impact of aerosol particles on low-level non- or slightly precipitating liquid water clouds. Some studies suggest that air pollution delays the onset of orographic precipitation whereas the effect of cities on

precipitation cannot be unambiguously related to air pollution (Levin and Cotton, 2009b).

Studies evaluating a possible aerosol effect on precipitation can be categorised among others into those that examine the weekly cycle of aerosols and precipitation, those analysing aerosol effects on precipitation from convective clouds and those from cyclones (extratropical cyclones and hurricanes). They can be summarised as follows:

Weekly cycles in cloud properties and precipitation have been reported as evidence for an aerosol effect on precipitation because of the weekly cycle in aerosol and their precursor emissions (e.g., Baeumer and Vogel, 2007; Rosenfeld and Bell, 2011). However many of these studies were disputed because of weaknesses in their applied statistical methods, their methodology or because the results could have been obtained by a simultaneous response of aerosol particles and precipitation to me-

teorological conditions (e.g., Sanchez-Lorenzo et al., 2012; Yuter et al., 2013; Boucher and Quaas, 2013).

Aerosol effects on convective clouds also found contradicting results. In the case of pyrocumulus which are characterised by extreme air pollution and extremely high aerosol concentrations Reutter et al. (2014) found that the onset of precipitation is delayed and its intensity is reduced with increas-

ing aerosol concentration. On the contrary, e.g. Rosenfeld et al. (2008) proposed an invigoration of convective clouds due to pollution. The growth of cloud droplets into rain drops is slower in polluted clouds, that consist of more but smaller cloud droplets, than in clean clouds. This delays the formation of warm rain and more cloud water is available for freezing in polluted deep convective clouds. The increased latent heat release may invigorate polluted convective clouds and lead to overall more

precipitation from them.

Igel et al. (2013) analysed the impact of aerosol particles on precipitation in the vicinity of a warm front. They found a shift in location caused by a delay of the onset of precipitation with increasing aerosol concentration, but the total precipitation from the warm front remained relatively constant. They attributed the rather constant total precipitation to a buffering effect. Smaller cloud

droplets due to increased aerosol concentrations in the mixed-phase region of the cloud, where cloud droplets and ice crystals co-exist, caused a decreased riming efficiency (Borys et al., 2003) but led to enhanced growth by diffusion. Thompson and Eidhammer (2014) studied the impact of aerosol particles on precipitation from a large winter cyclone. They also observed a delay in warm-phase precipitation formation, but an increase in snow to the north of the warm front. They concluded that

aerosol impacts were much stronger in areas with light precipitation rates than in those with higher precipitation rates.



The present study extends the above studies on the aerosol impact on one extratropical cyclone to a climatology of precipitation in extratropical cyclones in the North Pacific using the ECHAM6-HAM global climate model. We chose the North Pacific because of the increase in anthropogenic emissions over Southeast Asia and because the prevailing westerly winds carry these anthropogenic aerosols and their precursors over the North Pacific. Therefore we expect to see large differences in aerosol burden and aerosol optical depth between present-day and pre-industrial conditions in this region. In addition, low pressure systems frequently form here (e.g. Chen et al., 1991). Thus if an impact of anthropogenic aerosol particles on low pressure systems can be identified somewhere, then the North Pacific is the region to study this. The effect of Asian pollution on clouds in the North Pacific region has been investigated in different studies. Wang et al. (2014) showed that anthropogenic aerosols lead to changes in the cloud microphysical properties and the radiative forcing at the top of atmosphere. Furthermore they stated that an increase in cloud top height indicated invigorated mid-latitude cyclones connected to an overall increased precipitation. Also Zhou and Deng (2013) found that higher anthropogenic aerosol emissions lead to an increase in the amplitude of synoptic eddies and subsequently to an increase in surface precipitation. An intensification of the Pacific storm track is also found in Zhang et al. (2007). They argued that the wintertime Pacific is highly vulnerable to cloud-aerosol interactions because of the coupling between the Pacific storm track and Asian pollution outflow. In our study, we specifically focus on the so-called warm conveyor belt (WCB) airstream, which is a typical feature of (intense) extratropical cyclones. By focusing on the WCB we investigate aerosol effects in a relatively well-defined flow setting, in contrast to other climate model studies that consider the aerosol effect on total precipitation without distinguishing different categories of weather systems that produce precipitation (Denman et al., 2007). WCBs are coherent moist ascending airstreams in extratropical cyclones associated with the formation of elongated frontal cloud bands and intense precipitation (e.g. Browning, 1986). They can be objectively identified with the aid of trajectory calculations. Wernli and Davies (1997) showed that WCBs are formed by moist boundary layer air parcels that ascend by about 600 hPa or more within a time period of two days. Thereby intense cloud formation and latent heating occurs in WCB trajectories, leading to a typical loss of specific humidity of more than $10 \, \mathrm{g \, kg^{-1}}$ and an increase of potential temperature of about 20 K (Madonna et al., 2014). Initially, WCB air parcels are cloud free, then liquid water clouds form in the early part of the ascent leading to mixed-phase clouds in the middle troposphere and pure ice clouds in the WCB outflow at upper-tropospheric levels (about 350 hPa with temperatures below $-30°C$) (Joos and Wernli, 2012; Martinez-Alvarado et al., 2014). WCBs are intrinsic sub-synoptic scale features of extratropical cyclones and therefore climatological frequency maxima of WCBs occur in the extratropical storm track regions (Eckhardt et al., 2004; Madonna et al., 2014). In these regions more than half of total precipitation and up to 90% of extreme precipitation events are associated with WCBs (Pfahl et al., 2014). In particular in the western North Pacific just to the east of Japan more than 60% of the climatological precipitation and more than 90% of the pre-



cipitation extremes (defined as events above the 99th percentile) are collocated with WCBs (Pfahl
et al., 2014). Furthermore, due to the strong ascent, WCBs connect the different tropospheric layers
and are therefore important for the transport of pollution from the boundary layer to the mid/upper
troposphere (Stohl, 2001; Ding et al., 2009). By investigating precipitation formation in WCBs in
the western North Pacific, this study examines potential effects of anthropogenic aerosol emissions
within a highly relevant category of extratropical weather systems.

The paper is organised as follows. In section 2, the data and methods are introduced. Section 3 gives
an overview over the simulations and in section 4, a case studies of a WCBs is briefly discussed. In
section 5, the influence of aerosol particles on precipitation in WCBs is examined statistically for
the whole ten year climatology. The conclusions follow in section 6.

## 2    Data and Methods

### 2.1    ECHAM Simulations

The version of ECHAM6-HAM used in this study (ECHAM6.1-HAM2.2) has been described in
Neubauer et al. (2014). ECHAM6 (Stevens et al., 2013) solves prognostic equations for temperature,
surface pressure, divergence and vorticity in spectral space with a triangular truncation. ECHAM6
has a fractional cloud cover scheme that diagnoses fractional cloud cover from relative humidity
once a critical relative humidity is exceeded following Sundqvist et al. (1989). Differently from the
one-moment cloud microphysics scheme for stratiform clouds that is used in the standard model
ECHAM6, a two-moment cloud microphysics scheme is used in this study (Lohmann and Hoose,
2009). It consists of prognostic equations for the number and mass concentrations of cloud droplets
and ice crystals next to specific humidity.

The second version of the two-moment aerosol scheme Hamburg Aerosol Module (HAM2) pre-
dicts the aerosol mixing state in addition to the aerosol mass and number concentrations (Zhang et al.,
2012). The size-distribution is represented by a superposition of seven log-normal modes including
the major global aerosol compounds sulfate, black carbon, organic carbon, sea salt and mineral dust
in different mixing states. The latest version of HAM (HAM2.2) used here includes a size-dependent
in-cloud scavenging parameterisation (Croft et al., 2010). ECHAM6 with the two-moment cloud mi-
crophysics scheme is coupled to HAM by activation of aerosol particles with radii larger 35 nm into
cloud droplets (Lin and Leaitch, 1997), by homogeneous freezing of supercooled solution droplets
for the formation of cirrus clouds (Lohmann et al., 2008) and heterogeneous nucleation (immersion
freezing of internally mixed mineral dust and black carbon aerosols and contact freezing of exter-
nally mixed mineral dust particles) in mixed-phase clouds (Hoose et al., 2008). Thus, the impact of
aerosols on warm, mixed-phase and ice clouds can be studied using ECHAM6-HAM.

A mass flux scheme is employed for shallow, midlevel, and deep convection (Tiedtke, 1989) with
modifications for deep convection according to Nordeng (1994). The scheme is based on steady-



state equations for mass, heat, moisture, cloud water, and momentum for an ensemble of updrafts
and downdrafts, including turbulent and organised entrainment and detrainment. Detrainment of
cloud liquid water and ice in the upper part of the convective updrafts is used as a source term in
the stratiform cloud water equations. Aerosol effects on convective clouds are not included except
that the cloud droplet number concentration from detrainment from convective clouds depends on
the aerosol number concentration of internally mixed aerosol particles with radii larger 25 nm at the
cloud base of the convective clouds.

The ECHAM6-HAM simulations have been carried out in T63 horizontal resolution ($1.875° \times 1.875°$)
on 31 vertical levels with the model top at 10 hPa and a timestep of 12 minutes. All simulations used
present-day climatological sea surface temperature (average over the years 1979-2008) and sea-ice
extent and have been integrated for 10 years after a spin-up period of three months. The greenhouse
gas concentrations are constant and correspond to values of the year 2000. The present-day simu-
lations conducted with ECHAM6-HAM use aerosol emissions of sulfate, black and organic carbon
from the AeroCom Phase II data base for the year 2000 (Lamarque et al., 2010). Mineral dust and sea
salt emissions are calculated online based on near-surface wind speed. The sources of black carbon
aerosol particles are fossil fuel combustion, biofuel and wild fires. Only a fraction of the wild fires
is of natural origin, the rest of the emissions are due to anthropogenic activities. To isolate the total
anthropogenic aerosol effect, all simulations were repeated with aerosol emissions of sulfate, black
and organic carbon for pre-industrial times representative for the year 1850 (Lamarque et al., 2010).
The two simulations will be referred to as the present-day (PD) and the preindustrial (PI) simulation,
respectively.

**2.2  Calculation of WCBs**

In order to identify WCBs, trajectories are calculated with the Lagrangian Analysis Tool LAGRANTO
(Wernli and Davies, 1997) and the same procedure is used as in Madonna et al. (2014). Forward tra-
jectories are calculated for a time period of 48 hours, using wind fields from the ECHAM6-HAM
output every 6 hours. Trajectories are started every 6 hours during the entire simulation period (10
years). In the vertical, trajectories are started every 20 hPa in the lower troposphere between 1050 and
790 hPa, and horizontally, they are started every 150 km in the whole North Pacific (100 - 260 °E, 0
- 90°N). As mentioned in the introduction, only trajectories with an ascent of more than 600 hPa in
48 hours are selected as WCB trajectories. Additionally, WCB trajectories must rise in the vicinity
of an extratropical cyclone to distinguish them from, e.g., organised deep convection. Extratropical
cyclones have been identified using the method described in Wernli and Schwierz (2006). Therein, a
surface cyclone is defined as a local sea level pressure (SLP) minimum surrounded by the outermost
closed SLP contour. The area inside such a closed contour is then defined as an extratropical cyclone.
WCB trajectories have to cross the area of a surface cyclone at least once during their 48 h ascent.
For a more detailed description of the identification of WCB trajectories see Madonna et al. (2014).



The resulting climatology of WCB starting points is shown for both simulations in Fig. 1a,b. The distributions are similar in the PD and PI simulations and also agree well with the reference ERA-Interim based WCB climatology (Fig. 1c), in particular in the western and central North Pacific, which is the region of interest in this study.

For a more detailed analysis of the WCB trajectories (cf. section 5), different variables of interest
are traced along the trajectories. For our study this includes additionally to the position of the trajectories (longitude, latitude and pressure), the variables potential temperature ($\theta$), specific humidity ($q$), liquid water content (LWC), ice water content (IWC), cloud droplet number concentration (CDNC), precipitation (the sum of large scale and convective), black carbon (BC) aerosol mass mixing ratio and sulfur dioxide ($SO_2$). The BC mass mixing ratio is shown as a sum of all BC mass mixing ra-
tios in the soluble Aitken, accumulation and coarse modes. Please also note that the precipitation field is two-dimensional and therefore the interpolated precipitation value at the position of a WCB trajectory represents the precipitation reaching the surface below the trajectory, whereas all other variables are three-dimensional and the interpolated values represent the value of this field at the trajectory position itself. The selection of WCBs and the tracing of variables is performed for both
the PI and PD simulations.

### 2.3 Design of study

Our main focus is on studying WCBs that ascend in the western North Pacific, i.e. in a region where they can be potentially influenced by strong anthropogenic emissions. Most WCBs start their ascent over the ocean (Fig. 1), whereas emissions occur over the continent further to the west. This
constellation leads to a high variability of the concentration of anthropogenic pollutants ($SO_2$ and BC aerosols) in the inflow of North Pacific WCBs, as some WCBs contain only clean marine boundary layer air and others contain also highly polluted air parcels of continental origin. To cope with this variability, we quantified for every identified WCB trajectory the concentration of $SO_2$ at the start of the ascent and use this information to classify those 10% of the trajectories with the lowest (highest)
$SO_2$ concentration as clean (polluted) trajectories.

However, the cleanest and the most polluted WCBs in the western North Pacific tend to start at slightly different latitudes (not shown) and are therefore also characterized by slightly different initial specific humidity values, which renders a direct comparison of the evolution of the two categories of WCBs difficult. Since at the beginning of the ascent the cleanest WCBs are on average
moister than the most polluted ones, it would be impossible to attribute differences in the microphysical evolution along the two categories of WCBs to either the initial moisture value or the different aerosol concentrations. To circumvent this problem, we restricted the selection of WCBs to a relatively small area and to a narrow range of initial specific humidity values. The region considered for the start of the WCB ascent extends from 140-160°E and 20-40°N, which contains the climatolog-
ical maximum of WCB starting points in winter (see Fig. 1). In order to analyze WCBs that start




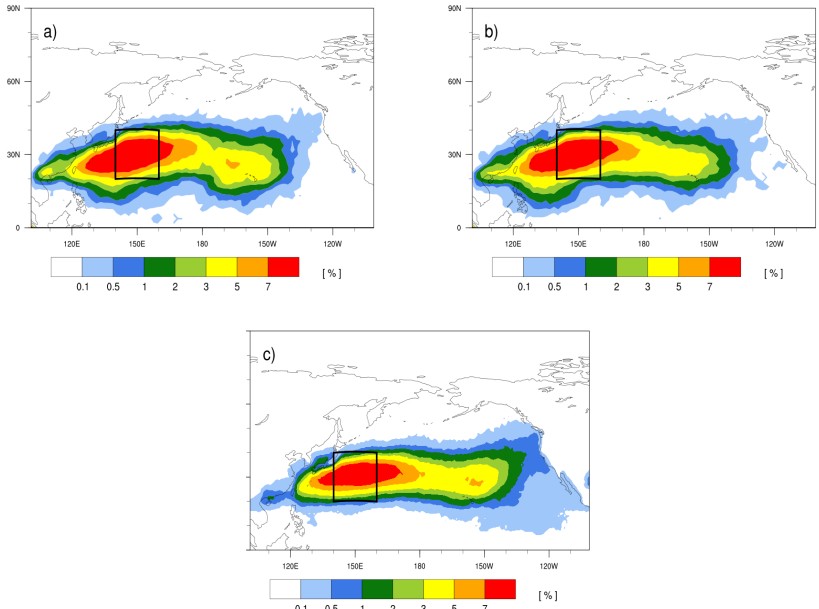

**Figure 1.** Climatological frequency of WCB starting points for PI (a), PD (b) and ERA-Interim (c). Colours represent the relative frequency (in %) of WCB trajectories at each grid point. The black box denotes the starting region of the WCBs considered in the analysis below.

with similar initial specific humidity values, only WCB trajectories with an initial specific humidity between 8-10 g kg$^{-1}$ will be considered for the statistical analysis. Thus, in the PD simulation, clean and polluted WCB trajectories are selected if they start their ascent in the box outlined above, their initial specific humidity value is between 8-10 g kg$^{-1}$ and if their SO$_2$ mass mixing ratio at time 0 h,

i.e. at the beginning of the ascent is below 51.6 pg kg$^{-1}$ or above 350.1 pg kg$^{-1}$, respectively, which corresponds to the 10% cleanest and the 10% most polluted WCB trajectories. In contrast, in the PI simulation, the 10% WCB trajectories have been selected whose initial SO$_2$ mass mixing ratio is closest to the mean (45-55 percentile). This selection procedure yields $\sim$ 2300 WCB trajectories in each category, which allows for a meaningful statistical analysis.

**3 Overview on pre-industrial and present day simulations**

Before evaluating the WCBs, some general characteristics of the ECHAM6-HAM simulations are presented and compared with ERA-Interim reanalyses (Dee et al., 2011). Figure 2(a-c) shows the winter mean field of SLP and potential temperature at 850 hPa for the PI and PD simulations and





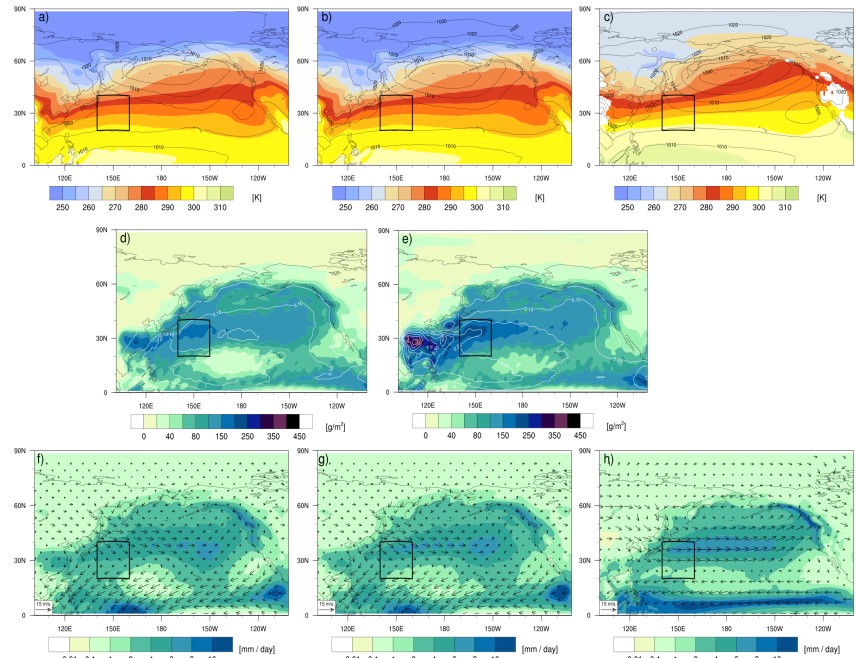

**Figure 2.** Time-mean SLP (contours) and potential temperature (shades) at 850 hPa for PI (a) and PD (b) simulations and ERA-Interim (c). Liquid water path (shading) in $g\,m^{-2}$ and aerosol optical depth (white contours) for PI (d) and PD (e) simulations. Total precipitation (large scale and convective in $mm\,day^{-1}$) and winds at 850 hPa in PI (f) and PD (g) simulations and ERA-Interim (h). The black box, extending from 140-160°E and 20-40°N, denotes the starting region of the considered WCBs.

ERA-Interim. The main features in the time mean are the general southward oriented gradient in SLP

and potential temperature, the Aleutian low with a central pressure of about 1000 hPa in the simulations and 995 hPa in ERA-Interim, and the strong horizontal temperature gradient in the region of Japan. The fields for the two simulations are almost identical, which reveals that the different aerosol emissions in the two simulations have no effect on the time mean SLP distribution and baroclinicity. Compared to ERA-Interim the ECHAM simulations underestimate the intensity of the Aleutian

Low, which is most likely due to the well known problem of fairly coarse global climate models to not resolve the peak intensity of extratropical cyclones (Jung et al., 2006). Another difference between the simulations and ERA-Interim appears for the tropical temperatures, which are about 5 K too low in the simulations which is consistent with the underestimation of tropical precipitation. This shortcoming of climate models can be related to the parameterization of tropical convection and can

be improved if the vertical resolution is increased (Stevens et al., 2013).



Marked differences between the two simulations occur when considering aerosol optical depth (AOD) and liquid water path (LWP, vertically integrated liquid water content) (see Fig. 2d, e). Per design, the PD simulation shows much higher values of AOD over China, Japan and the nearby oceanic regions than the PI simulation. In the WCB starting region defined in the previous section (black box in Fig. 2) the time-mean values of AOD vary roughly between 0.1 and 0.15 in the PI simulation and between 0.15 and 0.2 in the PD simulation, indicating that the mean AOD in the WCB starting region of interest is increased by less than 50% in PD compared to PI, but this increase is of course much larger over the main industrial areas. The increase in AOD is caused by an increase in accumulation and Aitken mode aerosol particles that serve as cloud condensation nuclei (CCN) and cause liquid water clouds to consist of more cloud droplets. Because the available water vapour remains the same, these cloud droplets do not grow as large. In a cloud consisting of more but smaller cloud droplets, the collision efficiency between cloud droplets is reduced and hence their growth to precipitation sized drops is retarded (Lohmann et al., 2016). In order to produce rain in a polluted cloud, the liquid water content needs to adjust to higher values. This is visible in the elevated values of the liquid water path in the PD simulation.

Considering again the WCB starting region, the increase of the LWP amounts to about 30%. Further downstream, the differences are small, illustrating the limited scale of the region impacted by the anthropogenic aerosol emissions. For the ice water path the two simulations show very similar mean values (not shown). Here no comparison is shown with ERA-Interim because different microphysical schemes lead to fairly large differences in liquid and ice water paths, which however cannot be interpreted as a model shortcoming. The ECHAM6-HAM microphysics is more sophisticated and complete compared to ERA-Interim and therefore, for these parameters, we cannot regard ERA-Interim as a reference.

Finally, Fig. 2f, g show the winter averaged surface precipitation and 850-hPa horizontal wind vectors for the two simulations and ERA-Interim (Fig. 2h) . Consistent with the differences discussed above, the simulations underestimate tropical convection in particular in the western Pacific, but north of 20°N the comparison with ERA-Interim shows only a weak underestimation. The lower values along the western flank of the Rocky Montains can be explained by the lower topography in the coarser resolution model. The lower tropospheric wind fields show the correct pattern in the simulations, however, in agreement with the errors in the SLP field, the westerlies in the main storm track region are too weak. Differences between the two simulations are generally small. The main differences appear for precipitation over China (where PI is slightly wetter) and in the central North Pacific (where PD has slightly more precipitation on average).



## 4 WCB case study

We first present an example WCB to illustrate the method and the evolution of $SO_2$ along the pathway of an initially strongly polluted WCB. Figure 3 shows a WCB identified with the method described above that occurred in the PD simulation in February. A total of 66 WCB trajectories have been selected in the western North Pacific, fulfilling the criteria of an ascent exceeding 600 hPa in 48 hours in the vicinity of an extratropical cyclone. The 66 WCB trajectories are shown as 66 in-

dividual lines in Fig. 3a and Fig. 3b indicates their fast ascent between times 0 and 48 h from, on average, 950 to 350 hPa. During their ascent the WCB air parcels move from their starting region to the east of Japan (35°N) to their outflow region over the central North Pacific (50-60°N, Fig. 3a). During the two days prior to their ascent, i.e., from time $-48$ to 0 h, the WCB trajectories are fairly stationary and experience, on average, a slow descent from 900 to 950 hPa (Fig. 3b). The $SO_2$ mass

mixing ratio (see colouring of trajectories) shows very high values for most of the trajectories before the ascent. A substantial fraction (46 of the 66 trajectories, i.e., 70%) exceeds the threshold to be classified as "polluted" (see section 2.3). During the ascent the $SO_2$ values rapidly decrease for three reasons: $SO_2$ can be oxidized to sulfate in the gas phase and subsequently serve as CCN, it can be dissolved in cloud droplets and oxidized in the aqueous phase (Seinfeld and Pandis, 1998), or it can

condense on other preexisting aerosol particles.

It is important to note that such a configuration where the majority of WCB trajectories are strongly polluted is rare (see further analysis below). More typically, a much smaller fraction of WCB trajectories is polluted due to the highly variable inflow of air parcels into a WCB.

## 5 Statistical analysis

In this section, first the general characteristics of the identified WCBs starting from the region defined above are described, followed by a detailed comparison of so-called clean and polluted WCBs in the simulations. It is our aim to compare, for PI WCBs and for clean and polluted PD WCBs, the evolution of LWC and IWC and the associated surface precipitation along the WCB trajectories in order to identify potential effects of the different initial aerosol concentrations on clouds and

precipitation.

In Figure 4, the averaged evolution of potential temperature, specific humidity, BC, CDNC, cloud condensate (sum of LWC and IWC) and surface precipitation is shown as a function of pressure, separately for the three categories of WCBs. The evolution of potential temperature and specific humidity (Fig. 4a,b) is very similar for the PD clean, PD polluted and PI WCBs. The trajectories

start their ascent between 290 and 295 K and reach the upper troposphere (i.e., 300 hPa) on the 315 K isentrope. The initial moisture is between 8 and 10 g kg$^{-1}$ (by design, see section 2.3) and decreases rapidly along the ascending trajectories at a very similar rate. This nicely shows that the overall meteorological conditions and the large-scale ascent of the WCB trajectories are comparable


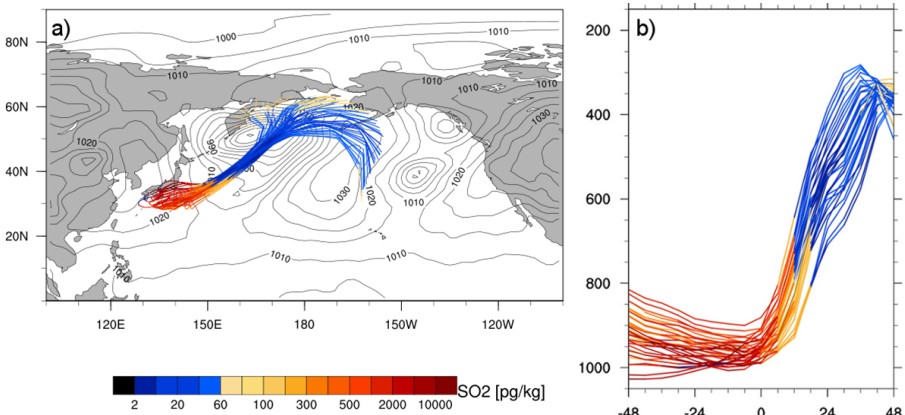

**Figure 3.** Case study of a WCB with 70% polluted trajectories calculated in the PD simulation. a) shows the path of the four-day WCB trajectories (plotted from time $-48$ h to $+48$ h, where 0 h denotes the start of the 48-h ascent), coloured with their $SO_2$ mass mixing ratio (in pg kg$^{-1}$) and SLP (black contours, in hPa) at time $+36$ h. b) The same WCB but showing pressure (in hPa) as a function of time (in hours), colored with the $SO_2$ mass mixing ratio.

between the three categories and should not be responsible for potential differences in the formation

of precipitation.

The evolution of BC (Fig. 4c) and $SO_2$ (not shown) along the considered WCB samples however reveals huge differences. The polluted trajectories exhibit high BC values at low levels with values between 50 and 130 pg kg$^{-1}$. Because the internally mixed BC aerosols serve as CCN and are activated to cloud droplets during the ascent, the BC mass mixing ratio strongly decreases with height.

The PD clean and PI trajectories show much lower BC values around 10-20 pg kg$^{-1}$ being five to ten times smaller than the polluted PD trajectories and even their 25-75% percentiles do not overlap below 750 hPa. The clean PD trajectories have twice as much BC as the ones in the PI simulation with a slight overlap of their 25-75% percentiles. This difference indicates that clean present-day conditions cannot be taken as a surrogate for pre-industrial conditions. For $SO_2$, PI and PD clean

trajectories show a very similar evolution whereas the polluted trajectories have much more $SO_2$ in the lower troposphere up to $\sim$700 hPa (not shown). The overall differences in $SO_2$ between the PD polluted and PI trajectories are even larger than for BC (Table 1).

The cloud droplet number concentration also decreases with decreasing pressure (Fig. 4d), partly because of the decrease of CCN with decreasing pressure and partly because the formation of pre-

cipitation increases with decreasing pressure due to the higher liquid and ice water contents (Fig. 4e) and the larger cloud droplets and ice crystals. The differences in CDNC between PD polluted and PD clean are much smaller than in BC partly because in ECHAM-HAM we assume that each cloud





**Figure 4.** Means (solid lines) and 25-75th percentiles (shades) of PD polluted (orange), PD clean (blue) and PI (black) WCB trajectories as a function of pressure for potential temperature (a), specific humidity (b), black carbon mass mixing ratio (c), cloud droplet number concentration (d), total condensate (sum of LWC and IWC) (e), and total surface precipitation (f).





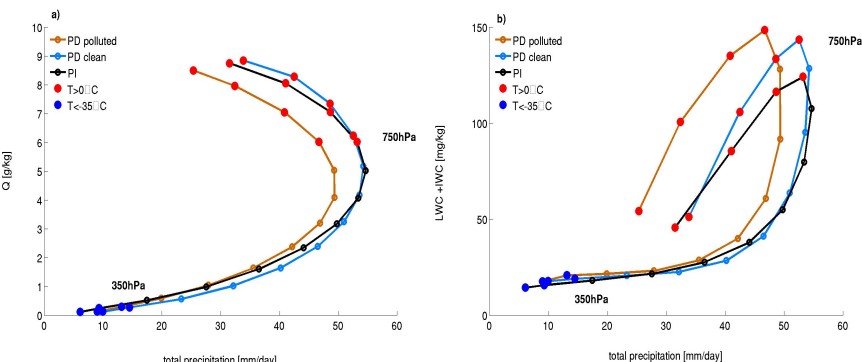

**Figure 5.** Phase diagram for the three categories of WCB trajectories: PD polluted (orange), PD clean (blue) and PI (black). (a) mean specific humidity (q) and (b) mean total condensate (LWC+IWC) as function of total precipitation; values are shown for 50-hPa thick vertical layers, from 900 to 300 hPa. The 750 and 350 hPa levels are marked and the red (blue) dots denote temperatures above 0°C (below -35°C).

**Table 1.** Different parameters averaged along all WCB trajectories for the PI, PD clean and PD polluted WCB trajectories.

| WCB | $q$ [g kg$^{-1}$] | conv. precip. [mm day$^{-1}$] | large-scale precip. [mm day$^{-1}$] | LWC [mg kg$^{-1}$] | IWC [mg kg$^{-1}$] | SO$_2$ [pg kg$^{-1}$] | BC [pg kg$^{-1}$] |
|---|---|---|---|---|---|---|---|
| PI | 3.7 | 2.4 | 17.6 | 29.4 | 5.4 | 23.2 | 9.1 |
| PD clean | 3.4 | 2.5 | 19.8 | 38.6 | 6.2 | 36.9 | 11.9 |
| PD polluted | 3.6 | 2.7 | 17.1 | 37.5 | 6.6 | 148.8 | 41.7 |

has a minimum CDNC of 40 cm$^{-3}$. Apart from that we see the same differences as in BC, with the PI trajectories having the smallest CDNC and the PD polluted trajectories the highest CDNC.

Going along with highest CDNC values in the PD polluted trajectories the median value of total condensate, i.e. the sum of the liquid and ice water content (LWC+IWC) (see Fig. 4e), is larger in the PD simulation, but the differences between the three sets of WCB trajectories are much smaller and the 25-75% percentiles strongly overlap. The total condensate peaks at 750 hPa in all simulations. At lower pressure levels the cloud condensate is reduced on the one hand due to precipitation formation

becoming more efficient higher in the cloud and on the other hand due to the smaller specific humidity at colder temperatures, i.e. smaller condensation/deposition rates. While the PD clean trajectories were in between the PI and PD polluted trajectories in terms of BC and CDNC, they now fall on top of the PD polluted ones. Note that the case-to-case variability in every sub-sample is larger than the systematic difference between them as to be expected due to the high variability in the associated

cyclone dynamics.





Precipitation formation depends more strongly on the liquid and ice water contents than inversely on the number concentrations of cloud droplets and ice crystals. Therefore its maximum at 700 hPa (Fig. 4f) is more determined by the maximum in liquid and ice water content than by CDNC and ice crystal number conventation. At 900 hPa the median precipitation rate is higher in the PD clean and

the PI trajectories than in the PD polluted trajectories, although the variability between individual trajectories is again rather large and the 25-75% percentiles strongly overlap. At pressures below 470 hPa, the median precipitation rate of the PD polluted trajectories crosses that of the PI trajectories, pointing to a delay in precipitation formation in the PD polluted WCBs. As summarised in Table 1 most of the precipitation falls as stratiform large-scale precipitation. It is largest in the PD clean

trajectories where the average specific humidity is smallest, i.e. where more water exists in the liquid and ice phase.

In order to investigate the effect of the differences in the aerosol loadings on the precipitation formation, phase space diagrams of $q$ and total condensate (LWC + IWC) as a function of the total precipitation are shown in Fig. 5.

The amount of condensate that can be formed is determined by the large-scale ascent, the availability of moisture and the conversion efficiency from condensate to precipitation. Fig. 5a, nicely shows that for all three cases the evolution of $q$ is very similar. All trajectories start with a specific humidity of $\sim 9$ g kg$^{-1}$ followed by a strong decrease at very similar rates. However, the precipitation falling out of these trajectories is reduced for the PD polluted subsample and the peak value

reaches less than 50 mm day$^{-1}$ in the PD polluted case whereas it reaches 55 mm day$^{-1}$ in the PD clean and PI case. This can also be seen in Fig. 5b, where the total condensate is shown as a function of total precipitation. However also the total amount of total condensate varies between PD and PI cases. With decreasing pressure, total condensate increases up to a height of 750 hPa. For the PD clean and PD polluted trajectories, this increase is steeper and the total condensate peaks around 140

mg kg$^{-1}$ whereas it increases more slowly and only reaches $\sim 120$ mg kg$^{-1}$ for the PI case.

Because of the much lower availability of CCN in the PI run (see Fig. 4c), fewer cloud droplets form that can grow to larger sizes. These larger and fewer cloud droplets form precipitation quite efficiently, leading to a fast removal of condensate from the atmosphere. On the contrary, already in the PD clean case, the mass mixing ratio of BC is twice as large as compared to PI. This increase

in the availability of CCN leads to more and smaller cloud droplets and reduces the efficiency of precipitation formation (see above). Therefore, less precipitation is reaching the ground and more condensate remains in the atmosphere. When even more CCNs are available as in the PD polluted subsample, the precipitation is more strongly reduced. This means that for the same liquid and ice water content, less can be converted to precipitation shifting the phase curve in the PD polluted sam-

ple to the left. An increase in liquid and ice water content due to a reduced precipitation formation can be observed when going from PI to PD clean conditions.





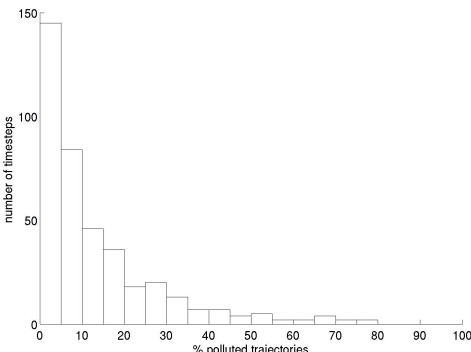

**Figure 6.** Histogram of the fraction of polluted WCB trajectories (in percent) as a function of the timesteps obtained from all days where more than 50 WCB trajectories start in the black box.

The difference in the onset of precipitation formation between PD clean and PD polluted trajectories could cause a spatial shift in the location of precipitation if a WCB consisted entirely of clean or polluted trajectories. This, however, is not the case as all WCBs have a large fraction of clean
trajectories (Figure 6). It can be seen that for the majority of timesteps where WCB trajectories start, less than 20% of all starting trajectories are classified as "polluted". Given that very few strongly polluted WCBs exist, it is not surprising that hardly any differences are observed in the DJF mean precipitation between pre-industrial and present-day conditions in Figure 2. In other words, the systematic effect of pollution observed in Fig. 5 is blurred by the high variability of inflow conditions
for WCBs consisting of typically about 50 WCB trajectories.

## 6  Conclusions

In this study we have analyzed the possible impact of aerosol particles on precipitation formation in WCBs of winter time extratropical cyclones in the North Pacific using the ECHAM6 global climate model coupled to the aerosol module HAM. We chose the North Pacific because here the difference
in aerosol burden and aerosol optical depth between present-day and pre-industrial times is among the largest due to the rise in emissions in Southeast Asia. Combined with the prevailing westerly winds, if differences in aerosol load upwind of the genesis regions of extratropical cyclones have an effect on them, we expect to see an effect in this region.

To investigate in detail the possible impact of aerosols on precipitation, we selected the most
polluted and cleanest trajectories occuring in the PD simulations. The polluted trajectories start with 5-10 times higher black carbon aerosol concentrations. The comparison between the most polluted and cleanest trajectories shows that for the most polluted cases, CDNC is clearly increased and the precipitation formation is reduced for a given total water content.



Our main findings are that despite these pronounced differences in the PD clean and PD polluted
trajectories, the overall amount of precipitation in the North Pacific has hardly changed between
pre-industrial and present-day conditions. This is due to the fact that all the analyzed WCBs have
both polluted and clean trajectories. Therefore precipitation formation inside the WCBs is initialized
first in the clean trajectories. This combined with the large variability of total condensate and total
precipitation in the different WCBs explains why no signal in precipitation in extratropical cyclones
due to anthropogenic aerosol particles can be detected. Our study confirms the findings by Igel et al.
(2013) for a single warm front where no change in overall precipitation was found. A shift in pre-
cipitation as found by Thompson and Eidhammer (2014) is not inconsistent with our results because
we showed that the amounts of precipitation are very variable within different WCB trajectories.

This study has however several caveats. First of all, it is a pure model study because no observa-
tional climatological data of the impact of aerosol particles on extratropical cyclones exist. Aerosol
impacts on clouds and precipitation are generally hard to detect in observational studies because
remote sensing studies suffer from not being able to detect aerosols and clouds simultaneously.
Moreover they present an Eulerian view whereas we analyzed the possible impact of aerosols on
WCBs in a Lagrangian way.

Another caveat is that WCBs are less well resolved in a global climate model than they are in
the regional model studies cited above. Given that the regional model study by Igel et al. (2013)
also did not find an effect on total precipitation, the resolution may not be a major issue. A more
important shortcoming could be that we used climatological sea surface temperatures which, to a
large extent, control the global mean evaporation and hence precipitation rates. Thus, changes in
the overall amount of precipitation could be larger if we had coupled the atmospheric GCM to a
mixed-layer ocean or a full dynamic ocean model.

*Acknowledgements.* EM and KW acknowledge support by the ETH Research Grant CH2-01 11-1.



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
