# Peer review of "Effect of anthropogenic aerosol emissions on precipitation in warm conveyor belts in the western North Pacific in winter – a model study with ECHAM6-HAM"

_Atmospheric Chemistry and Physics, 2016_

## Referee Comment (RC1) · Anonymous Referee #1 · 17 Nov 2016

Review of "Effect of anthropogenic aerosol emissions on precipitation in warm conveyor belts in the western North Pacific in winter – a model study with ECHAM6-HAM" by H. Joos et al.

This is one of the first studies to investigate the impact of aerosols on precipitation in extratropical cyclones in global models and extends the results of two previous cloud-resolving model studies that simulated just one ETC. As such, it is an important contribution to the field. In agreement with these previous two studies, the authors find no systematic change in precipitation due to aerosols despite large changes to cloud

droplet number concentration. The explanation for why there is no change in precipitation could be improved and a few inconsistencies exist in the discussion as detailed below. Overall this paper should be accepted with minor revisions.

Major Comments:

1. Lines 351-361. This paragraph does not seem to be well substantiated by the figures. "larger and fewer cloud droplets . . . lead to a fast removal of condensate from the atmosphere". However Figure 4e shows that there is virtually no difference between the clean and polluted trajectories in terms of condensate. The PI trajectories have less condensate throughout, and the rate of removal (after the peak at 750 hPa) is actually slower than for the PD trajectories. Then, the authors state that more BC in the PD clean trajectories compared to the PI trajectories reduces the efficiency of precipitation formation, yet Figure 4f shows the precipitation production to be nearly the same.

2. It took me a long time to understand what was being shown in Figure 6 and discussed in Lines 362-370. This description of the figure needs to be improved. Up until this point, "WCB" and "WCB trajectory" are used almost interchangeably which makes the phrase "all WCBs have a large fraction of clean trajectories" very confusing. I finally understood that here WCB is used to mean a collection of trajectories that all start at the same time. Perhaps the y-axis could say "number of WCBs"? Also, by definition, only 10% of trajectories are "polluted." So it is not surprising that a low fraction of the trajectories are polluted. I would expect that the corresponding plot for clean trajectories would look similar.

3. Line 386-388. These two sentences are unclear. Why does it follow that the precipitation is initialized first in the clean trajectories? Plus, as mentioned by the authors, even the clean PD trajectories are much more polluted than the PI trajectories. So why is there still such little difference in the precipitation? I don't think that this question has been addressed sufficiently.

4. A statement of data availability is missing.
* * *
Minor Comments:

5. In the abstract, the authors state, "We conclude that while polluted warm conveyor belt trajectories start with 5-10 times higher black carbon concentrations, the overall amount of precipitation is comparable in pre-industrial and present-day conditions. Precipitation formation is however suppressed in the most polluted warm conveyor belt trajectories." At first, these sentences seem contradictory. Without further information, one assumes that the present-day conditions are the same as the polluted WCBs. I recommend changing the first sentence to "While present-day conditions contain some polluted warm conveyor belts . . ." or something similar.

6. Line 362. The authors have not shown a difference in the onset of precipitation since all trajectories show precipitation occurring at the lowest levels.

7. Figure 4d. Does the mean really lay outside of the 25-75th percentile? It's theoretically possible, but seems unlikely.

8. For me, Figure 5, particularly 5a, does not add much insight to the discussion since it shows the same information as in Figure 4, just plotted differently.

Typos:

1. Line 9: suppressed

2. Line 253: mountains

3. Line 329: concentration

---

## Referee Comment (RC2) · Anonymous Referee #2 · 18 Nov 2016

Review of "Effect of anthropogenic aerosol emissions on precipitation in warm conveyor belts in the western North Pacific in winter - a model study with ECHAM6-HAM", by Joos et. al.

This paper examines the anthropogenic aerosol effects on the precipitation from warm conveyor belts (WCB) in the wintertime over North Pacific Ocean. The authors use the ECHAM6-HAM global climate model coupled to the aerosol module HAM, and the aerosol effects are represented by difference of the most polluted and cleanest trajectories of PD simulations. The authors find that the change in overall amount of precipitation due to anthropogenic aerosols is small and insignificant, but the precipitation is suppressed in most polluted warm conveyor belt trajectories.

However, the aerosol effects on precipitation only could be detected in the most polluted cases, and reliability of the statistical methods is also suspicious. Thus, the paper should be revised in response to the major criticisms and resubmitted.

**Major criticisms.**

The aerosol effects on warm conveyor belt is evaluated by the difference between the most polluted and the clean cases of PD simulations. The authors state that only trajectories has a very similar initial amount of moisture (8-10 g/kg) are compared, which ensures that differences in precipitation can be attributed to differences of the aerosol loading. Similar moisture provides similar initial condition for the precipitation, but it cannot ensure the precipitation to be the same. The precipitation is influenced by many other factors (e.g. atmospheric circulation, stability) besides humidity. One evidence is that the initial precipitation of PI simulation (none aerosol effects) exhibits large diversities (Figure 4f) with very similar initial humidity (Figure 4a). Thus, even with similar initial humidity, the difference of the polluted and the clean cases could not be simply attributed to the aerosol effects.

Moreover, even assuming that there is no aerosol effects in the model (a chemical transport model), such difference still could probably be detected in the simulation based on present statistical methods. It is because the most polluted cases could be correlated to the weak scavenging (less precipitation). The average precipitation of starting region of WCB is up to 8 mmd$^{-1}$, which may efficiently scavenge the aerosols from the atmosphere and reduce aerosol concentration. In such situation, the polluted (clean) cases could always accompany with less (more) precipitation. If comparing the most polluted cases with the cleanest cases, it could be possibly found that more aerosols tend to "reduce" the precipitation, although there are even no aerosol effects.

It is noticed the total condensate (LWC+IWC) difference is quite small between the polluted and clean cases (Figure 4e). It provides an evidence for that the difference of clean and polluted cases in precipitation could be not due to the aerosols. The aerosols tend to reduce the precipitation, through slowing down the autoconversion of cloud water to rain water with the increase of LWC. However, the differences of the LWC

between the polluted and clean cases are very small and insignificant. It implies that the difference in the precipitation may not be a result of the aerosol indirect effects.

The author states that no signal can actually be seen from comparing all WCBs in the PI simulation to all WCBs in the PD simulation, which is due to the very high variability of pollution inside one WCB. Aerosol effects can only be seen when comparing the cleanest with the most polluted trajectories. I agree with the author in this point. However, the polluted cases could be related to the less initial precipitation. Therefore, based on present methods, it is hard to tell whether the precipitation difference is a result or a cause for different cases. The only possible way to evaluate aerosol effects on the most polluted WCB is to compare the most polluted with the cleanest cases with similar initial precipitation, which is strongly recommended to be included in the revised manuscript. The manuscript could only be accepted in the situation that there is significant precipitation difference based on the recommended method.

**Other criticisms.**

Line 8: "supressed" should be "suppressed".

Line 66-74: Previous studies show that the anthropogenic aerosols tend to invigorate midlatitude cyclones and the related precipitation, which is contradictory to the results of this study. Corresponding explanations and discussions should be included in the manuscript.

Line 174: The concentration of sulfate aerosol should be compared. It because the AOD and CCN change are mostly from sulfate (Yan et al. 2015).

Line 195: Why are the cleanest WCBs on average moister than the most polluted ones? Do the authors check the relationship between aerosol concentration and initial precipitation?

Line 224: The precipitation from reanalysis data could still have some bias. The authors should better make a comparison with the observed precipitation (e.g. GPCP precipitation).

Line 230: A comparison of simulated (PD run) with observed (MODIS, MISR) AOD should be made here.

Line 240: The CCN, CDNC and LWP are significantly increased due to anthropogenic aerosols. Does it mean a strong second indirect aerosol effect? Why is the change of precipitation small and insignificant, while there is a dramatic cloud property change?

Line 245: How does aerosols affect ice water path in ECHAM6-HAM?

Line 246: Although different microphysical schemes lead to fairly large differences in liquid and ice water paths, a comparison of observed (MODIS) and simulated LWP is

strongly recommended to be made here. The reason is that the simulated LWP affects aerosol indirect effect and the readers should know such important information.

Line 265: I might miss something here. How are these cases selected from the 2300 cases? Are those 69 cases shown here only for an example? The authors should make it clear here.

Line 272: The authors state that only a small portion of WCB trajectories is polluted. Why is the number of polluted cases larger than that of clean ones? Just by coincidence? It is better to show the polluted and clean cases with the same number in two different panels.

Line 300: The sulfate concentration and the CCN number should be compared here, for they determine the CDNC and LWP change. The internal mixed BC could serve as CCN, but its contribution could be much smaller than sulfate and OC. I don't quite understand why the authors only choose BC for analysis.

Line 313: According to my opinion, the reason could be that CDNC change is determined by the concentration of sulfate and POM, other than BC.

Line 318: Compared to the PI simulation (clean and polluted together), the LWP of PD simulation is larger, which implies significant aerosol indirect effect. Is the precipitation also significantly reduced accordingly in PD (clean and polluted together) run? In the response to the quick report, the authors state that the precipitation changes for all cases (clean and polluted together) are insignificant. It seems that the WCB precipitation does not change much, although there is significant aerosol indirect effect.

Line 335: The total precipitation of PD polluted cases (19.8 mmd$^{-1}$) is very close to that of PI cases (20 mmd$^{-1}$), which is contradictory to the conclusions of the manuscript. The authors state that "Precipitation formation is however suppressed in the most polluted warm conveyor belt trajectories." In the most polluted cases, the total precipitation is almost the same as the precipitation of pre-industry time. Explanations should be given here.

Line 336: Without the uncertainty ranges, it cannot be concluded that the average humidity of PI clean case is the smallest.

Figure 5: At initial point (900hPa), the precipitation of PD polluted case is much smaller (by 10 mmd$^{-1}$) than that of PD clean and PI cases, while the total condensate is almost the same. How to explain such dramatic precipitation difference?

Line 353: Effective radius change should be given.

Line 353-355: Again, the CCN concentration should be given. It should be a standard output of the model. BC's contribution to CCN is quite small.

Line 357 and Line 362: For most levels, the precipitation of PD clean run is larger than that of PI run (Figure 4f and Figure 5b), which is contradictory to the authors' statements.

Line 359: Why is the LWC almost the same for PD clean and PD polluted cases, but there is significant difference in precipitation?

Line 366: Difficult to understand. What does "timestep" mean? How to get "20%" from figure 6?

**Tables and Figures.**

Table 1: uncertainty ranges should be given.

Table 1: CDNC number should be given.

Table 1: With this table, it is impossible to know whether the difference between cases is statistically significant. An additional table including the difference between three type of cases should be shown with uncertainty ranges. Meanwhile, the significant changes are shown in boldface.

Figure 2: For model evaluation, only the comparison of PD simulation and ERA-Interim should be included.

Figure 2: The authors state that the overall amount of precipitation is comparable, but it is difficult to get such information from figure 2. Thus, the AOD, LWP, precipitation and CDNC change between PD and PI simulation should be plotted in a separated figure with significance information (based on student t-test) to make the change issue clearer.

Figure 5: More vertical levels should be marked to make it clear.

**Reference:**

Yan H, Qian Y, Zhao C, et al. A new approach to modeling aerosol effects on East Asian climate: Parametric uncertainties associated with emissions, cloud microphysics, and their interactions[J]. Journal of Geophysical Research: Atmospheres, 2015, 120(17): 8905-8924.

---

## Author Comment (AC1) · 24 Mar 2017

Review of "Effect of anthropogenic aerosol emissions on precipitation in warm conveyor belts in the western North Pacific in winter – a model study with ECHAM6-HAM" by H. Joos et al.

This is one of the first studies to investigate the impact of aerosols on precipitation in extratropical cyclones in global models and extends the results of two previous cloudresolving model studies that simulated just one ETC. As such, it is an important contribution to the field. In agreement with these previous two studies, the authors find no systematic change in precipitation due to aerosols despite large changes to cloud droplet number concentration. The explanation for why there is no change in precipitation could be improved and a few inconsistencies exist in the discussion as detailed below. Overall this paper should be accepted with minor revisions.

Thank you very much for your comments to our manuscript. They helped us to improve the text and to make the main points better understandable.

Major Comments:

1. Lines 351-361. This paragraph does not seem to be well substantiated by the figures. "larger and fewer cloud droplets : : : lead to a fast removal of condensate from the atmosphere". However Figure 4e shows that there is virtually no difference between the clean and polluted trajectories in terms of condensate. The PI trajectories have less condensate throughout, and the rate of removal (after the peak at 750 hPa) is actually slower than for the PD trajectories. Then, the authors state that more BC in the PD clean trajectories compared to the PI trajectories reduces the efficiency of precipitation formation, yet Figure 4f shows the precipitation production to be nearly the same.

The effect which is described in these lines refers to Fig. 5b. We produced the phase space plots because in the evolution of variables along the ascending trajectories in Fig. 4, the signal of a precipitation reduction cannot be clearly seen.  In Fig. 4 we can mainly see that at a fixed pressure level, PD clean and polluted have higher total condensate values as PI and that PI and PD clean have higher precipitation rates. However, a direct link of total condensate and precipitation can only be seen in Fig. 5. There we can directly see that whenever the same total condensate values occur, the precipitation reaching the surface is reduced from PD polluted, to PD clean to PI. This is something which cannot be seen from Fig. 4. We think that the results of Fig. 4 and 5 are therefore not contradictory but complement each other. We added additional text to this paragraph (see lines 352,353 and 365-370)

2. It took me a long time to understand what was being shown in Figure 6 and discussed in Lines 362-370. This description of the figure needs to be improved. Up until this point, "WCB" and "WCB trajectory" are used almost interchangeably which makes the phrase "all WCBs have a large fraction of clean trajectories" very confusing. I finally understood that here WCB is used to mean a collection of trajectories that all start at the same time. Perhaps the y-axis could say "number of WCBs"? Also, by definition, only 10% of trajectories are "polluted." So it is not surprising that a low fraction of the trajectories are polluted. I would expect that the corresponding plot for clean trajectories would look similar.

We agree that Figure 6 and the associated description was not very clear. We changed the text (see lines 380-385).

A WCB consists of  a collection of trajectories whereas a varying fraction of the trajectories is classified as polluted.

It is true that only 10% of all the trajectories starting during the 10 year simulation are defined as polluted. However, it could also be that whenever a WCB starts, it is either completely clean or polluted, meaning that all the trajectories that belong to one WCB are either clean or polluted. This

would give a strong signal at 100% in Fig. 6. However, as this is not the case, it can be concluded that for the majority of times when a WCB starts, it consists of a mixture of clean and polluted trajectories, and typically only a small fraction is polluted (less than 30%).

3. Line 386-388. These two sentences are unclear. Why does it follow that the precipitation is initialized first in the clean trajectories? Plus, as mentioned by the authors, even the clean PD trajectories are much more polluted than the PI trajectories. So why is there still such little difference in the precipitation? I don't think that this question has been addressed sufficiently.

We see a clear impact of aerosols on the formation of precipitation when we compare the most polluted to the cleanest trajectories. However, as has been shown in Fig. 6, WCBs consist of a mixture of clean and polluted trajectories, whereas there are almost always much more clean trajectories in one WCB than polluted. This means that the effect of precipitation suppression in the most polluted WCB trajectories is damped by the more numerous clean trajectories in which no modification of precipitation occurs. Therefore, the overall effect on precipitation is very small and cannot be seen when looking at the whole North Pacific. We added some explanation (see lines 405-410).

4. A statement of data availability is missing.
Thanks, we included it in the manuscript.

Minor Comments:

5. In the abstract, the authors state, "We conclude that while polluted warm conveyor belt trajectories start with 5-10 times higher black carbon concentrations, the overall amount of precipitation is comparable in pre-industrial and present-day conditions. Precipitation formation is however suppressed in the most polluted warm conveyor belt trajectories." At first, these sentences seem contradictory. Without further information, one assumes that the present-day conditions are the same as the polluted WCBs. I recommend changing the first sentence to "While present-day conditions contain some polluted warm conveyor belts : : :" or something similar.

We extended the abstract and rewrote also he sentence you mention.

6. Line 362. The authors have not shown a difference in the onset of precipitation since all trajectories show precipitation occurring at the lowest levels.

We changed this sentence to "The slight delay in the precipitation formation between PD clean and PI...." (see line 378)

7. Figure 4d. Does the mean really lay outside of the 25-75th percentile? It's theoretically possible, but seems unlikely.

Yes, this is the case because the distribution is strongly skewed and there are some very high values of CDNC which shifts the mean to this high value.

8. For me, Figure 5, particularly 5a, does not add much insight to the discussion since it shows the same information as in Figure 4, just plotted differently.

As mentioned above (see comment 1), Fig. 5 enables the direct comparison of values of the same moisture or total condensate to the associated surface precipitation. We think that this figure is necessary to highlight the shift in precipitation for a given total condensate or moisture. We therefore decided to keep this figure.

Typos:

1. Line 9: suppressed done
2. Line 253: mountains done
3. Line 329: concentration done

---

## Author Comment (AC2) · 24 Mar 2017

Review of "Effect of anthropogenic aerosol emissions on precipitation in warm conveyor belts in the western North Pacific in winter - a model study with ECHAM6-HAM", by Joos et. al.

This paper examines the anthropogenic aerosol effects on the precipitation from warm conveyor belts (WCB) in the wintertime over North Pacific Ocean. The authors use the ECHAM6-HAM global climate model coupled to the aerosol module HAM, and the aerosol effects are represented by difference of the most polluted and cleanest trajectories of PD simulations. The authors find that the change in overall amount of precipitation due to anthropogenic aerosols is small and insignificant, but the precipitation is suppressed in most polluted warm conveyor belt trajectories. However, the aerosol effects on precipitation only could be detected in the most polluted cases, and reliability of the statistical methods is also suspicious. Thus, the paper should be revised in response to the major criticisms and resubmitted.

**Major criticisms.**

The aerosol effects on warm conveyor belt is evaluated by the difference between the most polluted and the clean cases of PD simulations. The authors state that only trajectories has a very similar initial amount of moisture (8-10 g/kg) are compared, which ensures that differences in precipitation can be attributed to differences of the aerosol loading. Similar moisture provides similar initial condition for the precipitation, but it cannot ensure the precipitation to be the same. The precipitation is influenced by many other factors (e.g. atmospheric circulation, stability) besides humidity. One evidence is that the initial precipitation of PI simulation (none aerosol effects) exhibits large diversities (Figure 4f) with very similar initial humidity (Figure 4a). Thus, even with similar initial humidity, the difference of the polluted and the clean cases could not be simply attributed to the aerosol effects.

Please note that the PI simulation also includes aerosol effects. The difference to the PD runs is only in the amount of aerosol emissions.

Moreover, even assuming that there is no aerosol effects in the model (a chemical transport model), such difference still could probably be detected in the simulation based on present statistical methods. It is because the most polluted cases could be correlated to the weak scavenging (less precipitation). The average precipitation of starting region of WCB is up to 8 mmd$_{-1}$, which may efficiently scavenge the aerosols from the atmosphere and reduce aerosol concentration. In such situation, the polluted (clean) cases could always accompany with less (more) precipitation. If comparing the most polluted cases with the cleanest cases, it could be possibly found that more aerosols tend to "reduce" the precipitation, although there are even no aerosol effects.

Thank you very much for your remark. We agree that the amount of pollution in the WCB inflow, thus at the initial timestep can be influenced by the amount of precipitation. A high precipitation rate could lead to enhanced scavenging of aerosols and thus to a reduced aerosol load in the WCB inflow and vice versa. However, we think that this fact does not change the interpretation of our results due to the following reason:
The amount of precipitation falling at the same point in time as the start of the WCB might influence the initial aerosol concentration of the WCB trajectory. However, the WCB starts its ascent with the initial aerosol concentration which is shown in Fig. 4c), no matter which process determined this concentration. In the following, the air parcels start their ascent along the WCB trajectories and start to form LWC, IWC and precipitation. The formation of

the cloud condensate and the associated precipitation formation is then determined by the initial aerosol concentration, independent of the process which was responsible for creating the aerosol concentration at the WCB inflow. After that we follow the ascending air parcel with its given aerosol concentration and determine the formation of clouds and precipitation in a Lagrangian framework. We think that the difference between the aerosol concentration in the WCB inflow and the cloud and precipitation formation along the ascending airstreams, thus the Lagrangian perspective, was not well enough explained in the original manuscript. We therefore added more explanations in the text (see lines 344-350).

As suggested by the reviewer below, we did an additional analysis in order to investigate whether the difference in the initial aerosol concentration is formed due to more/less precipitation at the beginning of the trajectories. We therefore selected from the trajectories that start with a similar Q the ones that have similar precipitation at the starting time. Thus, we selected only trajectories that have an initial precipitation between 5 and 15 mm/day (the results do not depend on the exact values which are selected). Please note that in Fig. 4f, the precipitation is plotted as a function of pressure and the precipitation at time t=0h can't be seen from this figure. This selection yields between 360 and 430 trajectories for the different subsamples PD polluted, PD clean and PI. The results can be seen in Figure 1:

[Figure]

Fig. 1: Evolution along WCB trajectories as a function of pressure of black carbon mixing ratio (c), total condensate (sum of LWC and IWC) (e) and total surface precipitation (f) for trajectories starting with the same initial precipitation (between 5 and 15 mm/day) selected from the trajectories starting with the same initial moisture.

Very similar to Figure 4 in the paper, the polluted trajectories (orange subsample) have a much higher BC mixing ratio even if the initial precipitation in the subsamples is the same. Also the evolution of total condensate (e) and precipitation (f) does not show a significant difference to the trajectories shown in Figure 4 of the paper. We therefore conclude that it is not the initial precipitation that determines the amount of pollution at the start of the trajectories. The polluted trajectories tend to come more from the west (Japan) as compared to the clean trajectories.

It is noticed the total condensate (LWC+IWC) difference is quite small between the polluted and clean cases (Figure 4e). It provides an evidence for that the difference of clean and polluted cases in precipitation could be not due to the aerosols. The aerosols tend to reduce the precipitation, through slowing down the autoconversion of cloud water to rain water with the increase of LWC. However, the differences of the LWC between the polluted and clean cases are very small and insignificant. It implies that the difference in the precipitation may not be a result of the aerosol indirect effects.

Yes you are right, the differences are quite small. The strongest aerosol signal might be the delay in precipitation formation. However, we also state in our manuscript that the case-to-case variability  in every sub-sample (PI, PD polluted, PD clean) is larger than the

systematic difference between them. This high variability is expected as there is a very high case-to-case variability in the associated cyclone dynamics. However, we think that a notable signal can be seen which shows that the suppression of the precipitation formation leads to an enhanced amount of condensate in the atmosphere. This can be seen in Fig. 4f, where the PD polluted line is below the two others and also in Fig. 5b. There it can be seen that going from PI to PD polluted, the precipitation reaching the ground is reduced (see the shift of the brown curve to the left) and the amount of total condensate is increased (compare peak values of all three curves). Although this signal is not extremely pronounced we still think that we can understand and explain the observed patterns and relate it to the aerosol concentration.

The author states that no signal can actually be seen from comparing all WCBs in the PI simulation to all WCBs in the PD simulation, which is due to the very high variability of pollution inside one WCB. Aerosol effects can only be seen when comparing the cleanest with the most polluted trajectories. I agree with the author in this point. However, the polluted cases could be related to the less initial precipitation. Therefore, based on present methods, it is hard to tell whether the precipitation difference is a result or a cause for different cases. The only possible way to evaluate aerosol effects on the most polluted WCB is to compare the most polluted with the cleanest cases with similar initial precipitation, which is strongly recommended to be included in the revised manuscript. The manuscript could only be accepted in the situation that there is significant precipitation difference based on the recommended method.

As stated above, the processes determining the initial aerosol concentration in the WCB inflow are not important for the following cloud and precipitation formation in a Lagrangian framework and the suggested additional analysis clearly shows that initial aerosol concentration differences are independent of initial precipitation (see Fig.1, above).

**Other criticisms.**

Line 8: "supressed" should be "suppressed". done

Line 66-74: Previous studies show that the anthropogenic aerosols tend to invigorate midlatitude cyclones and the related precipitation, which is contradictory to the results of this study. Corresponding explanations and discussions should be included in the manuscript.

We included discussions of the following papers ( Wang et al., 2014a, PNAS, Wang et al., Nature comm., 2014b) (see lines 414 - 433)

In Wang et al. (2014a) and Wang et al. (2014b), an influence of Asian pollution on cloud properties is also seen. As in our study, a strong increase in CDNC is simulated when going from PI clean to PD polluted conditions. The associated increase in LWP caused by a reduction of the formation of precipitation can also be seen in our study. However, the methods for investigating the aerosol effect on precipitation in extra-tropical cyclones strongly differ between our study and the studies presented by Wang et al (2014a,b). In Wang et al. (2014a,b) the impact of anthropogenic aerosols on the circulation and the storm track is investigated by comparing simulations with different aerosol loadings. In our approach, we use a feature based method in order to only investigate the impact on precipitation formation along the strongly ascending WCB trajectories. The Lagrangian approach following the air parcels enables us to directly asses the impact of aerosols on

the most cloud producing air stream in extratropical cyclones separately to other precipitating cloud systems.

As shown in Pfahl et al., 2014, WCBs lead to ~60 % of the total precipitation in the North Pacific. Our results suggest that precipitation formation in the WCB trajectories is not strongly influenced by aerosols. However, we can't make any statement about the precipitation formation in other cloud systems which are not as strongly dynamically driven as a WCB. Therefore, our results are not necessarily contradictory to Wang et al. (2014a, b). Also the fact that we use another global climate model and that the microphysical parametrization is a large source of uncertainty explains why the impact of aerosols on clouds and precipitation is not directly comparable between Wang et al. (2014a,b) and our study.

Line 174: The concentration of sulfate aerosol should be compared. It because the AOD and CCN change are mostly from sulfate (Yan et al. 2015).

We chose an aerosol species that is not produced inside clouds (like sulfate aerosols), which would have complicated the analysis. However please note, that the selection of air parcels as polluted/clean is done based on SO2 which is a precursor for sulfate aerosols. We also compared the evolution of SO2 and BC along trajectories and found a very similar behavior with a strong decrease with decreasing pressure (see line 305 ff)

Line 195: Why are the cleanest WCBs on average moister than the most polluted ones? Do the authors check the relationship between aerosol concentration and initial precipitation?

The cleanest WCB are moister because they start at a slightly different latitudes. The relationship between aerosol concentration and initial precipitation has been investigated (see comment above).

Line 224: The precipitation from reanalysis data could still have some bias. The authors should better make a comparison with the observed precipitation (e.g. GPCP precipitation).

The aim of this study is not to compare the model results of total precipitation to satellite data but instead to apply a Lagrangian technique in order to investigate how an increased aerosol concentration affects the formation of precipitation in WCBs. We therefore decided to keep the comparison to ERA-interim.

Line 230: A comparison of simulated (PD run) with observed (MODIS, MISR) AOD should be made here.

The main idea of this paper is not the validation of AOD compared to satellites, we decided to not include an additional comparison to satellite data. We think that our analysis and results would not benefit from such a comparison.

Line 240: The CCN, CDNC and LWP are significantly increased due to anthropogenic aerosols. Does it mean a strong second indirect aerosol effect? Why is the change of precipitation small and insignificant, while there is a dramatic cloud property change?

Due to the increase in CCN and CDNC, thus the first indirect aerosol effect, the collision efficiency is reduced and the growth to precipitation sized drops is retarded. In order to produce rain in a polluted cloud, the liquid water content needs to adjust to higher values.

However, the amount of precipitation in the annual global mean is balanced by the global mean evaporation rate. Because we use prescribed sea surface temperatures, the averaged changes in evaporation and precipitation is small (see text in lines 235-243).

Line 245: How does aerosols affect ice water path in ECHAM6-HAM?

Dust and black carbon act as ice nucleating particles.

Line 246: Although different microphysical schemes lead to fairly large differences in liquid and ice water paths, a comparison of observed (MODIS) and simulated LWP is strongly recommended to be made here. The reason is that the simulated LWP affects aerosol indirect effect and the readers should know such important information.

In Neubauer et al. (2014), the ECHAM6-HAM2 cloud scheme has been validated and compared to ERA-interim data as well as satellite data from CALIPSO and CERES products.

Neubauer, D., U. Lohmann, C.Hoose and M.G. Frontoso, 2014, Atmos. Chem. Phys., 14, 11997–12022.

Line 265: I might miss something here. How are these cases selected from the 2300 cases? Are those 69 cases shown here only for an example? The authors should make it clear here.

Fig. 3 shows one example of a WCB which consists of 66 trajectories. The WCB started at a certain point in time and each of the trajectories ascends at least 600hPa in 48h in the vicinity of a cyclone. Otherwise it would not have been selected. Thus, Fig. 3 does not show 66 cases but 66 trajectories belonging to one and the same WCB.
We added an additional explanation (see line 267-270 and line 282).

Line 272: The authors state that only a small portion of WCB trajectories is polluted. Why is the number of polluted cases larger than that of clean ones? Just by coincidence? It is better to show the polluted and clean cases with the same number in two different panels.

Fig. 3 is only one example. In this example ~70% of the trajectories in this WCB are classified as polluted. However, the statement that only a small portion of WCBs is polluted refers to all WCBs starting during the whole 10 years. In Fig. 6 it can be seen that for the majority of timesteps when a WCB (consisting of at least 50 trajectories) starts, then less than 20% of the trajectories belonging to one WCB are classified as polluted.

We agree that the description and use of the words "WCB" and "trajectory" was confusing. We changed the terminology in several places in the text.

Line 300: The sulfate concentration and the CCN number should be compared here, for they determine the CDNC and LWP change. The internal mixed BC could serve as CCN, but its contribution could be much smaller than sulfate and OC. I don't quite understand why the authors only choose BC for analysis.

We chose an aerosol species that is directly emitted and not only produced inside cloud droplets.

Line 313: According to my opinion, the reason could be that CDNC change is determined by the concentration of sulfate and POM, other than BC.

We agree that only part of the aerosols that serve as CCN are BC and that other compounds like sulfate aerosols are also very good CCNs. Thus, the much smaller increase in CDNC compared to the strong increase in BC from PI to PD might be caused by the fact that other compounds serve as CCN. However, we decided to look at BC because it is emitted directly as aerosol. We added a small remark to the text (see line 319)

Line 318: Compared to the PI simulation (clean and polluted together), the LWP of PD simulation is larger, which implies significant aerosol indirect effect. Is the precipitation also significantly reduced accordingly in PD (clean and polluted together) run? In the response to the quick report, the authors state that the precipitation changes for all cases (clean and polluted together) are insignificant. It seems that the WCB precipitation does not change much, although there is significant aerosol indirect effect.

We think it is important to note that one WCB consists of many trajectories and in the simulations we have many WCBs (each consisting of many trajectories).

Here we refer to Fig. 4e. It can be seen that the PD trajectories have the highest amount of total condensate (4e) but the smallest amount of precipitation (4f). However, in Fig. 4 we compare the most polluted and the cleanest trajectories. The statement "precipitation changes are insignificant" refers to the fact that in reality, WCBs do not consist of only polluted or clean trajectories but of a mix of both of them (see Fig. 6). Thus, the main statement is that when a direct comparison of the cleanest and most polluted trajectories is done, then an effect on precipitation can be seen (see also Fig. 5). But when we compare all precipitation related to WCBs then this effect is smeared out because WCBs always consist of polluted and clean trajectories. As also mentioned above, we checked again the whole text concerning the use of the words trajectory and WCB as this was misleading.

Line 335: The total precipitation of PD polluted cases (19.8 mmd-1) is very close to that of PI cases (20 mmd-1), which is contradictory to the conclusions of the manuscript. The authors state that "Precipitation formation is however suppressed in the most polluted warm conveyor belt trajectories." In the most polluted cases, the total precipitation is almost the same as the precipitation of pre-industry time. Explanations should be given here.

The precipitation is largest in the PD clean case (19.8 mm/d) and smaller in the PD polluted (17.1 mm/d) and PI (17.6 mm/d) cases. The effect of a reduced precipitation can be clearly seen in the mean and percentiles values when going from the PD clean to PD polluted cases. For the PI simulation, differences in the aerosol emissions lead to complex feedbacks in the model which leads to slight differences in the Meteorology between the two different simulations and to the low value in precipitation. However, the main conclusion of the manuscript is clearly supported by the numbers shown in table 2.

Line 336: Without the uncertainty ranges, it cannot be concluded that the average humidity of PI clean case is the smallest.

The averaged humidity of PI is not the smallest. It is 3.7 g/kg for PI and 3.4 g/kg for PD clean and 3.6 g/kg for PD polluted. We also added the 25 and 75 percentiles to the table for each variable.

Figure 5: At initial point (900hPa), the precipitation of PD polluted case is much smaller (by 10 mmd$_{-1}$) than that of PD clean and PI cases, while the total condensate is almost the same. How to explain such dramatic precipitation difference?

As the trajectories do not start with the same initial precipitation it is not possible to directly link the total condensate and precipitation already at 900 hPa. However, when looking at the evolution of total condensate along the ascending trajectories, differences can be seen. The total condensate increases with decreasing pressure up to ~750 hPa. This increase is steepest for the PD polluted case as there the formation of precipitation is suppressed most due to the higher CDNC and therefore more total condensate stays in the atmosphere. In PI, the removal of total condensate is more efficient leading to increased precipitation.

Line 353: Effective radius change should be given.

We think that the effective radius would not give additional information because the signal in the enhanced CDNC is very clear.

Line 353-355: Again, the CCN concentration should be given. It should be a standard output of the model. BC's contribution to CCN is quite small.

See statement above.

Line 357 and Line 362: For most levels, the precipitation of PD clean run is larger than that of PI run (Figure 4f and Figure 5b), which is contradictory to the authors' statements.

Thanks for this hint: We changed this sentence to: In the PD polluted subsample, where even more CCNs are available, the precipitation is more strongly reduced.

Line 359: Why is the LWC almost the same for PD clean and PD polluted cases, but there is significant difference in precipitation?

See comment to Fig. 5, above

Line 366: Difficult to understand. What does "timestep" mean? How to get "20%" from figure 6?

We agree that this part of the text was difficult to understand and we rewrote this part (lines 380-385).

**Tables and Figures.**

Table 1: uncertainty ranges should be given. We inlcuded the 25th and 75th percentiles.
Table 1: CDNC number should be given. We also included CDNC
Table 1: With this table, it is impossible to know whether the difference between cases is statistically significant. An additional table including the difference between three type of cases should be shown with uncertainty ranges. Meanwhile, the significant changes are shown in boldface.

Unfortunately it is not clear to us what should be done here. We do not have an ensemble of simulations which would allow us to determine statistical significance However, as suggested we inserted the percentile values for each variable along the trajectories in each sub-sample in order to highlight if the observed changes are only visible in the mean of each variable or also in the 25th and 75th percentiles (see Table 2).

Figure 2: For model evaluation, only the comparison of PD simulation and ERAInterim should be included.

We decided to keep Figure 2 as it is. The reason is that we are not primarily interested in a model evaluation but we also want to show the differences between the different model simulations PI and PD.

Figure 2: The authors state that the overall amount of precipitation is comparable, but it is difficult to get such information from figure 2. Thus, the AOD, LWP, precipitation and CDNC change between PD and PI simulation should be plotted in a separated figure with significance information (based on student t-test) to make the change issue clearer.

We included a table with the averaged values of LPW, precipitation, CDNC and AOD over our domain (North Pacific).  See table 1 and text in lines 259-264.

Figure 5: More vertical levels should be marked to make it clear. done

Reference:
Yan H, Qian Y, Zhao C, et al. A new approach to modeling aerosol effects on East Asian climate: Parametric uncertainties associated with emissions, cloud microphysics, and their interactions[J]. Journal of Geophysical Research: Atmospheres, 2015, 120(17): 8905-8924.